# The Role of Self-Efficacy, Work-Related Autonomy and Work-Family Conflict on Employee’s Stress Level during Home-Based Remote Work in Germany

**DOI:** 10.3390/ijerph19094955

**Published:** 2022-04-19

**Authors:** Martin Lange, Ina Kayser

**Affiliations:** 1Department of Fitness & Health, IST University of Applied Sciences, Erkrather Straße 220a-c, 40233 Düsseldorf, Germany; 2Department of Communication & Business, IST University of Applied Sciences, Erkrather Straße 220a-c, 40233 Düsseldorf, Germany; ikayser@ist-hochschule.de

**Keywords:** self-efficacy, work-related stress, self-perceived health status, remote work, work-family conflict, autonomy

## Abstract

Home-based remote work becomes increasingly popular. The facets of remote work, especially working from home, are multifaceted and can become stressors that affect a person’s health. At the same time, self-efficacy is an important personal resource to deal with health-related stressors. The objective of this study is therefore to explore the relationship between self-efficacy (SE), work-related stress (WRS), health outcomes (health and anxiety), contributing factors (autonomy and experience) and work-family conflict (WFC) in a remote work setting. Using a PLS-model (partial least square) with a sample of *n* = 5163 responses, we found that SE significantly reduces WRS (β = −0.164; *p* < 0.001). Moreover, WFC increases WRS and anxiety, while SE reduces WFC and mediates health outcomes (anxiety: β = −0.065; *p* < 0.001; health: β = −0.048; *p* < 0.001). At the same time, autonomy (β = 0.260; *p* < 0.001) and experience (β = 0.215; *p* < 0.001) increase SE. Our results have high practical implications for employers and employees, underlining the importance of self-efficacy as a personal resource to buffer WRS and WFC while promoting overall health at the same time.

## 1. Introduction

Remote work takes place outside a designated work location, such as corporate offices, and is often associated with working from home or working (home-based remote work) at a client’s location [1]. In contrast, telework can be conducted remotely as well, but also focuses on the use of personal electronical devices [2]. While home-based remote work is not entirely new, the number of businesses implementing working-from-home models grows rapidly. During the past five years, working remotely has become a promising working format for many businesses. In Germany, four out of ten companies offer a remote work model, in particular working from home, with an increasing tendency [3]. In 2018, about 5.1% of German employees worked remotely for at least half of their working hours. A total of 11.8% of Germans worked from home at least once a month. During the pandemic and especially during pandemic-related lockdowns of corporate offices, the proportion of employees working from home ranged from 16% to 30% in Germany [3,4]. This represents an increase of more than 200% to nearly 500% compared to pre-pandemic levels. As dealing with the pandemic becomes more of a routine, more companies permanently included remote work in their work setup [3,4]. Previous research shows that home-based remote work during the pandemic has different effects on an employee’s self-perceived stress and health. Moreover, prior studies suggest that self-efficacy can be a personal resource that reduces stress and promotes a person’s health. However, little is known about the relationship between self-efficacy and work-related stress (WRS) as well as overall health of employees in this specific context. The goal of this study is therefore to close this gap by analyzing the role of self-efficacy and associated constructs.

### 1.1. Facets of Home-Based Remote Work and Job Stress

Despite the beneficial aspects of home-based remote work, previous literature suggests that working from home is associated with a higher level of WRS [5,6,7]. WRS is defined as a transactional construct, which describes stress as a direct product of the transaction between an individual and the environment impacting one’s resources and wellbeing [8,9,10]. Contrastingly, other authors present previous research results indicating that working from home is linked to many positive aspects such as the reconciliation of work and family life due to flexible working hours, increased productivity of employees, higher job satisfaction and many more [11,12]. During the COVID-19 pandemic, home-based remote work was inevitable as an effective preventive measure to reduce infections [4]. Regardless of a pandemic situation and out of a corporate perspective, there are several economic aspects in favor of remote work such as lower rent expenses, lower maintenance costs of office space, less business-related travel and an increase in productivity amongst employees [11]. Employees benefit from reduced distractions in the office realm by working from home. This especially applies to office workers whose execution of work tasks must be concentrated, focused and knowledge-based. The possibility of working remotely from home is also associated with greater job satisfaction and commitment amongst employees [13,14].

On the contrary, most jobs—especially office workers—require a certain level of social interaction with superiors and/or coworkers. Working from home isolates employees physically from each other, which can be perceived as challenging for work-related activities carried out within teams [15]. The physical isolation and digital work execution thus limit the ease of addressing colleagues and superiors [13,14,16]. This claim is supported by previous research that shows that remotely working individuals felt isolated and rely on office interactions for social support [17,18,19]. Moreover, social interaction between coworkers and superiors is an important promoter of employee engagement and mental well-being [11].

A significant pillar of the ongoing discussion on the benefits of remote work is employee stress. While home-based remote work may lead to lower stress levels due to decreased commuting times and day-to-day office demands [13,14], it is at the same time linked to an “always-on” debate. This may lead to a higher number of working hours, difficulties to switch off from work [20], and to a perceived need of being constantly available [12]. These behaviors are usually set as a cultural norm in remote work settings by superiors and coworkers and can lead to an overall poor well-being and mental health problems [21,22].

### 1.2. Job Demand-Resource Model as a Framework to Analyze Resources and Demands

To approach the relationship between working from home and employee health systematically, the Job Demand-Resource model (JDR) offers a comprehensively evaluated framework [23]. The JDR is a well-established and recognized model in previous literature for WRS. It offers the advantage of incorporating personal resources from a broader perspective, which is in turn helpful to explore health-related relationships [24]. The JDR focuses on the interaction of two specific sets of factors leading to WRS or employee well-being: job demands and job resources [25,26]. Job demands can be physical, psychological, social or organizational demands, requiring a permanent psychophysical effort, and are—if high—linked to higher psychophysical costs [23]. In the context of remote work, job demands can for example be permanent noise, high work pressure, a high number of phone calls and video conferences that take up important work time, high-frequent interactions with demanding clients on the phone, interruptions by family members or inadequate IT equipment [11]. Job demands are not necessarily negative in nature, but can evolve as stressors [27]. Contrastingly, job resources can be physical, psychological, social or organizational aspects that are necessary to achieve work goals, buffer the effects of job demands and are connected to personal and professional development [23]. Ideally, job demands and job resources behave in a dynamic, balanced manner that lead to or maintain employee health. However, when work demands exceed work resources and work resources are inadequate for work demands, an imbalance occurs that leads to WRS over time.

The aspects mentioned above suggest that working remotely affects employees in different ways. Previous studies suggest that certain factors such as remote work experience, autonomy and self-efficacy can promote health-related outcomes and the overall well-being during remote work. Therefore, we aim to contribute to the ongoing discussion by examining the influence of such facilitating factors as well as the perception of stress in home-based work. Since the COVID-19 pandemic is increasing the amount of remote work, these health-promoting factors could be a key element to keep employees’ stress levels low.

As presented, previous research shows strongly heterogeneous effects of working from home on perceived job stress and health status. The JDR offers a framework that underlines the importance of various factors associated with employee health in the context of remote work. Self-efficacy can be a key element to reduce job stress and improve employees’ health. Furthermore, autonomy and remote work experience are two concepts that can promote self-efficacy impacted by work-family conflicts. Against this background, the aims of our investigation are
(1)to test the hypothesized role of self-efficacy and(2)to assess the strength of the relationships between self-efficacy, work-related stress and work-related outcomes in a German remote work population.

### 1.3. Association between Work-Related Stress and Health Outcomes

The relationship between WRS and health has been extensively evaluated in the past. Studies show that a high level of WRS is inversely linked to health-associated outcomes [23,28,29]. In the context of remote work, evidence of an effect of WRS remains controversial. Remote work decreases psychological and physiological stress when it comes to reducing commuting times, increased flexibility, productivity or an improved balance of private and work life [30]. Simultaneously, previous studies report an increased perception of psychological WRS associated with presenteeism [30], work-family conflict [31], social isolation and declining health behaviors such as physical activity during after-work hours [28]. Additionally, the association between WRS and health is predominantly associated with psychological symptoms such as fatigue [29,32], anxiety or depression disorders [28]. Based on the presented evidence, it can be postulated that: 

**Hypothesis** **1** **(H1).**
*WRS is negatively associated with overall health.*


### 1.4. Persistent WRS Leads to Anxiety and Depressive Disorders 

If WRS exceeds a certain level and persists over a longer period of time, job anxiety can arise [33]. Job-related anxiety is a person’s response to WRS, where the stressor is either persistent or overwhelming to a person. Employees with job anxiety see themselves in a situation unable to deal with job demands such as heavy workload, long working hours, job insecurity or difficulties with co-workers. Studies reveal that employees with job anxiety or depressive symptoms show a significantly higher level of WRS [34,35,36,37] and a ’psychologically impaired well-being’ [38]. Research in the context of remote work also suggests that job anxiety is not only favored by WRS, but also by a combination with private stress [35,39,40,41,42]. Contrary to the effects of WRS on overall health, the evidence leads to the hypothesis that:

**Hypothesis** **2** **(H2).**
*WRS is positively associated with job anxiety.*


### 1.5. Self-Efficacy Is a Central Factor That Reduces WRS 

Personal resources are a key element of dealing with WRS during remote work. Resilience, control or competence-oriented constructs such as self-efficacy are considered important personal resources in the JDR framework [27]. Self-efficacy has evolved from Bandura’s social cognitive theory and is defined as ’a judgment of one’s ability to execute a particular behavior pattern’ [43]. The underlying mechanism is the perception of being in control when encountering a potential stress-related situation, which functions as an important buffer. This is particularly important because people with higher self-efficacy tend to take on more challenging tasks, exert more effort and show more perseverance in these tasks. [44]. Furthermore, previous studies found that self-efficacious employees are more likely to show an increased level of confidence in executing new tasks, handling new situations with a positive attitude, and they are more likely to succeed in these novel tasks [44]. In some cases, the relationship between self-efficacy and WRS has to be regarded independently, especially when challenging tasks become obstacles beyond a person’s individual control, such as caring for others. In these cases, a U-shaped relationship can emerge [45]. Evidence on the inverse relationship between self-efficacy and WRS is well established [36,37], but has not been extensively investigated in a remote work setting. Applied to the context of remote work, job demands, and thus WRS, tend to be even higher due to the cumulation of multiple stressors such as longer working hours, higher workloads or difficulties caused by the overlapping of work and private life. Other findings underline the formation of the hypothesis [24] that in a remote work context:

**Hypothesis** **3** **(H3).**
*Self-efficacy is negatively associated with WRS.*


### 1.6. Positive Mediating Effects on Health Associated Outcomes 

The importance of self-efficacy increases when looking at health-associated outcomes. Self-efficacious employees tend to perceive less stress and feel in control of job demands. With people being at home, the support of supervisors or company resources is limited, as they cannot affect a person’s private life or certain events at an employee’s home. Employees have to manage their tasks mostly on their own and at the same time cope with the demands of their private life in addition to the professional ones. It is therefore an important skill to keep WRS at a manageable level. This, in turn, prevents prolonged stress episodes and the emergence of health-endangering risks. The lowering effect of self-efficacy on WRS leads to a higher job satisfaction, better health scores [46,47,48,49] and well-being [35]. At the same time, anxiety and depressive disorders are observed significantly less frequently with higher self-efficacy [50,51]. Employees with higher levels of job anxiety report high levels of job stress and low levels of self-efficacy [36,37,51]. Thus, it can be proposed that self-efficacy can be a crucial personal antecedent that is negatively related to WRS which in turn mediates the relationship between SE with overall health and job anxiety. Therefore, we postulate that:

**Hypothesis** **4** **(H4).**
*WRS mediates the relationship between SE and health-related outcomes such as job anxiety (H4a) and overall health (H4b).*


### 1.7. Remote Work Experience and Autonomy Promote Self-Efficacy

Given the inverse relationship between self-efficacy and WRS, the question arises as to which personal resources promote self-efficacy. Two concepts are directly linked to the level of self-efficacy, which are remote work experience and autonomy. Bandura [44] describes experience as one of the most crucial sources of self-efficacy alongside vicarious experience, verbal persuasion and physiological and affective states. When a person successfully completes a task, it generally has a direct positive impact on their self-esteem. A person then feels more confident to master similar tasks successfully, and self-efficacy improves. Several studies report a strong, positive relationship between work experience and self-efficacy; unfortunately, most of them were conducted in traditional work environments and not home-based remote work settings [52,53].

Self-efficacy gives people the feeling that they can successfully solve tasks and challenges on their own. This, in turn, requires an appropriate framework of autonomy. In the context of remote work, methodological autonomy and decision-making autonomy are particularly important. Decision-making autonomy means that an employee can independently make decisions that lead to solving a problem, whereas work-method autonomy gives employees the power to decide how they want to solve a problem. Autonomy in general is positively associated with self-efficacy [54], which is also supported by various other studies in different occupational contexts [55,56,57]. Positive associations between job autonomy and certain indicators of psychological well-being were also reported in previous literature [58]. Not only is autonomy a ’powerful motivational tool’ [54], it also gives an employee the necessary independence to deal with heterogeneous situations or problems. Contrary to that relationship, limiting a person’s autonomy can lead to lower job satisfaction and higher job stress. The aspects of autonomy and self-efficacy were mainly considered against the background of work design in terms of motivation and performance [59], but less in the context of health. Therefore, the concepts of experience and autonomy have huge potential in promoting self-efficacy in a remote work setting. Against this background, we postulate that self-efficacy is positively associated with work-method autonomy and decision-making autonomy. 

**Hypothesis** **5** **(H5).**
*Remote work experience is positively related to self-efficacy.*


**Hypothesis** **6** **(H6).**
*Work autonomy is positively related to self-efficacy.*


**Hypothesis** **7** **(H7).**
*Decision autonomy is positively related to self-efficacy.*


### 1.8. Work-Family Conflict Is a Circumstantial Factor Influencing Self-Efficacy and WRS

Lastly, the increasing blending of work and private life raises the issue of work affecting family life (work-family conflict) and, conversely, how family affects work (family-work conflict). Work-family conflict emerges when pressure in the work role prevents a person from meeting the demands of the private situation [24]. In the past, the way work affects family life was studied extensively, with particular attention to how shift work, overtime or constant accessibility affect workers’ personal lives [42]. Work-family conflict is associated with a strong gender dependency, as in most cultures women are predominantly responsible for housework and caregiving [60]. However, the understanding of work-family conflict has shifted significantly due to two major aspects. First of all, the growing acceptance of remote work as a main working concept leads to a two-way phenomenon that includes work-family conflict and family-work conflict at the same time [61]. Working from home eliminates a local distance between work and family life, leading to distractions, interruptions, sharing a workspace, double responsibilities and burdens such as preparing lunch for the family or helping with homework [62]. Second, working from home is related to a higher workload and longer working hours [12,63], leaving less time for family and other personal life activities [12]. Regardless of whether remote work is voluntarily chosen or coerced, both aspects underscore that remote work involves a number of factors that impact work and family life [24]. Evidence from previous literature clearly supports a connection to various health-related outcomes such as work-related stress, work-related depressive disorders including burnout, as well as life and marital dissatisfaction [31,62,64]. High levels of work-family conflict are linked to a higher overall level of perceived stress, physical fatigue and psychological weariness [31,62]. Hobfoll’s [65] ’Conversation-of-Resources’ theory explains that work-family conflict ties up important cognitive, emotional and physical resources, which are then no longer available for work-related performance. Over time, this loss of resource will lead to increased anxiety, psychological strain and health problems [33]. Based on the argumentation, we postulate that work-family conflict is impacting health-related outcomes negatively.

**Hypothesis** **8** **(H8).**
*Work-family conflict is positively related to WRS.*


**Hypothesis** **9** **(H9).**
*Work-family conflict is positively related to job anxiety.*


**Hypothesis** **10** **(H10).**
*Work-family conflict is negatively related to overall health.*


### 1.9. Self-Efficacy Mediates the Effects of Work-Family Conflict on Work-Related Stress

As described above, self-efficacy is considered an important personal resource that enhances general resistance to stress, is related to higher levels of resilience and promotes positive coping strategies with environmental demands [66]. Compared to these demands, self-efficacious individuals show a high sense of control and self-evaluation which lead to less stress in general [66,67,68,69]. This mechanism, in turn, reduces the levels of work-family conflict and WRS. The mediating effect of self-efficacy is shown in different studies, but predominantly outside the context of remote work [46,65,70,71]. Against this background, we postulate that self-efficacy is inversely associated with work-family conflict. 

**Hypothesis** **11** **(H11).**
*Work-family conflict is negatively related to self-efficacy.*


The conceptual model is presented in Figure 1. 

## 2. Materials and Methods

### 2.1. Design and Data Collection

We applied a cross-sectional survey design to gather data to test our hypotheses. In so doing, we collected nationwide data from mid-January to mid-February 2021 with the assistance of a panel service provider. The survey coincided with a period when companies in Germany were legally required to allow their employees to work from home if the circumstances of the job permitted this. With the assistance of the panel provider, participants were invited by email and provided with a survey link. We furthermore informed participants about the survey itself without referring to any research goals to avoid bias along with an informed consent and information about data safety measures according to the general data protection regulation (GDPR).

### 2.2. Sample Selection and Characteristics

Participants were randomly recruited by the panel service provider from a basic population that matched the following criteria: Individuals had to be employed fully or part-time (at least 50% of full-time working hours), work predominantly at a desk and work remotely at home with at least 50% of the full-time working hours. In addition to these inclusion criteria, the respondents were stratified with respect to demographic characteristics such as gender and age, regional distribution and educational level to obtain valid results for the structure of the German labor force.

We received *n* = 5163 valid responses. The mean age of the respondents is 44 years with a SD of 11 years. A total of 53.1% of the sample were female, and 46.9% were male. A total of 31.5% respondents were employed parttime. About a quarter of the sample had a middle school degree (23.8%), 25.6% a high school degree; about half of the sample had a higher degree of education (undergraduate 14.3%, graduate 30.3%, doctorate 2.2%), and 3.8% reported other degrees. 

### 2.3. Measures

With regard to the hypotheses presented, work-related stress, self-efficacy, work-autonomy, decision-autonomy, remote work experience, work-family conflict, self-perceived health status and job anxiety remain the constructs of interest. Additional variables were surveyed to control for personal, organizational and technological characteristics (gender, age, highest degree, years of employment at current job and size of the company).

To measure the selected constructs, we used validated instruments that are established in previous literature. The survey items were translated carefully following validated and broadly accepted guidelines [72], and the particular wording was adapted to the context of remote work. Three subscales with a total of 16 items were derived from Staple’s [53] remote work questionnaire to evaluate work-related stress (five items), self-efficacy (eight items) and remote work experience (three items). An example of the items used is ‘I work under a great deal of tension’. Staple’s questionnaire is compiled by validated scales [73,74,75] and is an appropriate measure that is used to predict relationships between the antecedents of remote work self-efficacy and its consequences. 

Work-method autonomy (three items) and decision-making autonomy (three items) were measured using subscales from the Work-Design Questionnaire by Morgeson, Humphrey [57]. The questionnaire was developed to assess work characteristics that are linked to job satisfaction in a remote context. Participants were asked for examples to assess how much ‘the job allows me to make a lot of decisions on my own’.

To assess job anxiety, we used the 10-item subscale focusing on job-related worries of the job anxiety scale by Muschalla, Linden, Olbrich [76]. The job anxiety scale is a validated research questionnaire that aims to measure different qualities of work-related anxiety. An example item is ‘I am always worrying about minor matters in my work and during all the working day’. All six constructs were measured with a 5-point Likert scale anchoring from strongly disagree (1) to strongly agree (5). 

Work-family conflict was measured with the work-family conflict subscale (five items) by Netemeyer, Boles, McMurrian [61] with a 5-point Likert scale ranging from strongly disagree (1) to strongly agree (5). The work-family conflict subscale assesses the self-perceived conflict between the work and family role. Participants had to respond to items such as ‘The demands of my work interfere with my home and family life’.

The Minimum European Health Module Item 1 (MEHM1) was used to measure self-perceived overall health on a 5-point scale anchoring from 1 = very good to 5 = very bad [77].

### 2.4. Data Analysis

We employed partial least squares-based structured equation modeling (PLS-SEM) to analyze our data using Smart PLS v. 3 [78]. The main reason to choose a PLS approach over covariance-based SEM is the complex structure of the model and the focus on the identification of key driving constructs [79].

## 3. Results

### 3.1. Measurement Model

To evaluate the reliability of each construct, we calculated both the Cronbach alpha value and the composite reliability (CR) for each construct (see Table 1). CR values and Cronbach’s alpha of each construct exceeded the recommended threshold of 0.70 for both measures [80,81]. To test for sufficient discriminant validity of the constructs, i.e., the extent to which each measure of the model is distinct from other variables, we assessed the average variance extracted (AVE) [81].

As shown in Table 1, each construct revealed an acceptable AVE value above the 0.50 [80] point to satisfactory convergent validity. Moreover, to evaluate discriminant validity, we used the strict Fornell–Larcker criterion [80]. We compared the square root of the AVE values with the correlations between the constructs. As displayed in Table 2, we found the square root of every AVE value on the diagonal to be larger than any correlation among latent constructs, indicating satisfactory discriminant validity of the model [80]. Since some correlations between constructs are high, we investigated the variance inflation factor (VIF) to assess the possible presence of multicollinearity in the independent variables. Multicollinearity could lead to the use of redundant information in the model [82] and thus to an inflated variance of predictors [80,83]. The variance inflation factor (VIF) analysis indicates a reasonable collinearity of the indicators, as all VIF values remain under the threshold 3.0 [80]; hence, multicollinearity is not a concern in our data set.

### 3.2. Structural Model

Table 3 summarizes the results of the structural model analysis. To evaluate the structural model, we assessed the path coefficient estimates and their significance levels. Most paths show significant results, indicating strong support for our hypotheses. The only insignificant path is work-family conflict on health. Therefore, H10 (work-family conflict is positively related to overall health) is not supported. 

Stress (WRS) revealed a direct and significant effect on health (β = 0.319; t = 16.282; *p* < 0.001). As high values of health resemble a bad health status, high values of WRS are associated with a lower health status supporting H1. Therefore, it can be assumed that working remotely from home is associated with higher perceived WRS levels leading to poorer overall health. 

Moreover, WRS shows a direct effect on job anxiety. The effect is positive and statistically significant (β = 0.396; t = 24.278; *p* < 0.001) supporting H2. At this point, high levels of WRS are linked to depressive and anxiety symptoms strengthening H1.

We applied a partial mediation following recommendations of Carrión, Nitzl, Roldán [84]. Self-efficacy revealed a direct effect on WRS (β = −0.164; t = 13.989; *p* < 0.001) as well as indirect effects on job anxiety (β = −0.065; t = 11.743; *p* < 0.001) and health (β= −0.048; t = 9.954; *p* < 0.001). All effects are statistically significant and negative, supporting the hypothesis that higher levels of self-efficacy reduce stress and therefore improve health associated outcomes. H4a and H4b are hence supported.

We moreover found direct effects of the three self-efficacy supporting factors remote work experience, work autonomy and decision autonomy. Remote work autonomy (β = 0.215; t = 18.784; *p* < 0.001), work autonomy (β = 0.153; t = 8.672; *p* < 0.001) and decision autonomy (β = 0.260; t = 15.049; *p* < 0.001) are positively and statistically significant linked to self-efficacy, lending support to H5, H6 and H7. The identified evidence underlines the importance of experience and autonomy to manage challenges during remote work.

Regarding the concept of work-family conflict, most of the proposed hypotheses could be supported. The direct association between work-family conflict and WRS (β = 0.634; t = 62.723; *p* < 0.001) as well as job anxiety (β = 0.254; t = 14.912; *p* < 0.001) were statistically significant and positive. H8 and H9 are thus supported. It can be assumed that home-based remote work has an impact on private life, leading to increased perceived stress (WRS) and anxiety symptoms. The opposite assumption of a lower work-family conflict resulting in a better state of health could not be verified by the results of our study (β = −0.017; t = 0.889; *p* < 0.374), leaving H10 without empirical support. Lastly, self-efficacy and work-family conflict are related in a negative way (β = −0.281; t = 22.964; *p* < 0.001), supporting H11.

The f^2^ values were assessed to evaluate effect sizes of our research model. Values ranging from 0.020 to 0.150, 0.150 to 0.350 or larger or equal to 0.350 indicate weak, medium or large effect sizes, respectively [85]. Following this classification, most hypotheses show weak effect sizes, whereas the f^2^-value for H8 indicates a large effect size (Table 3).

To evaluate the overall fit of the estimated model, we used the standardized root mean square (SRMR) [86]. Other model fit indices that are commonly used with variance-based path SEM are less common in the context of covariance-based PLS path analyses and are controversial in the literature with regard to their informative value [87]. The overall fit of the estimated model (SRMR = 0.065; d_ULS_ = 1.385; d_G_ = 0.313) is satisfactory. Models with a SRMR below the cut-off value of 0.080 are considered acceptable [86]. 

The coefficient of determination (R^2^) and predictive relevance (Q^2)^ for the dependent variables are displayed in Table 4. R^2^ values account for the amount of variance of the latent endogenous variable that is being explained by the antecedent exogenous variable. Chin [88] states that R^2^ values of 0.67 show a substantial, of 0.33 moderate and of 0.19 weak explanation of variance. Therefore, the explained variance in overall health is weak in our model, whereas the other latent endogenous variables show a moderate explanation of the variance. Q^2^ is a measure of predictive relevance, i.e., how well the empirical data can be reconstructed using the model [88]. A Q^2^-value greater than zero indicates predictive relevance for the respective construct [89,90] which is given in the present study.

## 4. Discussion

The significance of self-efficacy and work-related stress has been extensively studied in traditional workplace environments and with a focus on productivity in recent decades [43,44,48,91,92,93]. Our goal was contrastingly to analyze this relationship within the context of home-based remote work and selected self-efficacy promoting factors out of a health-oriented perspective. Firstly, the findings underline that work-related stress is strongly associated with health-related outcomes and confirm the importance of self-efficacy in balancing stress levels. Secondly, the results expand the scope of this relationship by pointing out the role of work-family conflict as an important circumstantial factor and the role of autonomy as a work-related resource. To maximize the benefits of home-based remote work in terms of performance [94] and health, it is important to reduce prolonged periods of work-related stress and promote self-efficacy enhancing resources. 

Within an office environment, an employer can create an atmosphere that promotes performance and supports employee health by regulating office space, reducing distractions and noise, physically supporting colleagues or providing optimal technical equipment. The same is true for employees. The level, as well as the amount, of support decreases dramatically when working from home. In fact, most circumstantial factors regarding remote work are not the responsibility of the employer, as they can neither mandate nor control how to set up and maintain a home-based work setting. Some of these circumstantial factors cannot even be influenced by employees either, such as taking care of elderly persons or children. Personal resources such as self-efficacy as well as work conditions such as autonomy are therefore becoming increasingly important. The results of our study revealed positive effects of self-efficacy on both constructs and are in line with previous literature leading to the original hypotheses.

Despite restrictions on workplace design at home, employers can create health promoting work conditions or supply employees with health enhancing resources. Autonomy can be both an employee’s resource and a work condition [54,95,96]. In this study, we evaluated work and decision-making autonomy, which both showed significant effects on self-efficacy. As for the path coefficients and effect sizes, decision autonomy showed a stronger association than work autonomy. At the time of data collection, the COVID-related lockdown and legally required home-based remote work may have restricted work autonomy if working hours and availability were fixed to ensure appropriate business procedures. Moreover, the behavior of superiors also affects the degree of autonomy, which was not assessed in this study.

Additionally, remote work experience turned out to be another factor that employers should consider to promote self-efficacy. Giving employees the option to work remotely from home has multiple implications. First, it promotes an employee’s self-efficacy, since they have to work mostly alone and manage individual workloads by themselves. Second, it lets employees experience work and family life at regular and self-dosed intervals that fit the personal demands of the employees, preventing sudden mental overloads and stress [52,97,98].

However, the results do not reflect on the full extent of the relationship between self-efficacy and work-family conflict. Cultural and gender-related aspects could interfere with the work role, if women are more involved in housework or childcare [60]. Thus, remote work can in turn can lead to higher stress levels for certain groups.

Overall, the results underline most of our hypotheses that self-efficacy is a key construct to balance the stress level of a person, as well as circumstantial factors such as interference with work and family life. The only hypothesis not supported by the results was H10, which postulated an inverse relationship between work-family conflict and overall health. The relationship between work-family conflict and any health outcome in general is pointed out by various studies [24,31,33,64]. Although our results confirmed this hypothesized relationship between work-family conflict and job anxiety, work-family conflict and health did not show any relevant association. This result might be caused by the single item measure we used to assess overall health. Though this particular single item measure is well established in the literature [77] and may actually prevent mindless response behavior [99], single-item measures can also be a cause of concern due to possible lower reliability, content validity and sensitivity [100]. Another reason for this striking result could be the short time span to reach a measurable change of perceived health.

### 4.1. Practical Implications

The presented study focused on important aspects of home-based remote work with high practical implications for self-efficacy and work-related stress. A positive and healthy organizational culture combined with an empowering leaderships style [101] builds a necessary foundation for home-based remote work [11]. Employers need to accept that home-based remote work settings are different, have heterogeneous impacts on employees and that employees’ remote work settings are not manageable for employers. Therefore, the degree of control is limited for supervisors, who must adjust accordingly regarding work design and leadership style [102,103]. Human-resource development should focus on managers, supervisors and employees at the same time with training on autonomous working, giving and receiving individual feedbacks and the creation of individual solutions. Managers and supervisors should participate in trainings to learn how to detect stress-related factors and how to provide support to employees with such demands. Some employees do not have an ideal work set-up at home and need different types of support from their employers, e.g. flexible working hours [58]. Moreover, giving employees an option to work remotely from home promotes a trustworthy organizational culture and lets employees deal with challenges of work- and family life on a self-selected pace.

### 4.2. Limitations

The present study has various limitations. The cross-sectional design will not allow any insight into the duration or variation of work-related stress levels over time. It is not clear for how long high stress levels are perceived before leading to anxiety symptoms or poorer health. Moreover, short-term circumstantial factors might distort the results, which could affect individuals substantially regarding workload, sudden family or work events. Furthermore, self-reported measures are well established in social science, yet they come with certain disadvantages regarding accuracy, construct validity and cultural differences [104]. Shortly before data collection commenced, a federal law on remote work was enacted by the German government, which could increase workers’ perceptions of work-related stress compared to a deliberate remote work situation. At the same time, our results reflect a more realistic situation for that specific period of time. Similar to a previous study [28], we did not assess working hours, which could lead to a distortion regarding the interference of work with family life and work-related stress. The employees’ working hours could range from 20 to 40 h or more per week of home-based remote work. Shorter working hours could intensify work and increase perceived stress. Longer working hours could also have a greater impact on work-family life and indirectly cause a higher perceived stress level.

### 4.3. Future Research

Future research should focus on additional aspects that promote personal resources of employees, such as working conditions, interventions and targeted group-specific aspects. Previous studies suggest that an independent leadership style amplifies the influence on self-efficacy positively [101] especially when employees experience success completing a task, receive positive feedback from supervisors or colleagues or work in an emotional-supportive work surrounding. Organizational culture and leadership styles are important sources of self-efficacy [44]. Therefore, impacts of different types of leadership or organizational culture on self-efficacy for home-based remote work should be a focus of future research. Moreover, using another framework other than the JD-R might be useful in identifying other factors by applying different perspectives such as Conservation of Resources theory. In this study, we did not take a closer look at work strains or other resources [105]. Future studies may also analyze interventions that promote self-efficacy for specific target groups such as supervisors and managers or women and men, since the present study did not apply a socio-demographic differentiation of the sample. The present study assessed constructs of mental health and general health. Future research may analyze various physical constructs of health. Self-efficacy is strongly associated with health behaviors such as physical activity [106] that reflect directly to different constructs of physical health which in turn support aspects of mental health [107].

## 5. Conclusions

Home-based remote work is an important work environment to which employers and employees have to adapt. Despite the beneficial aspects of working from home, our results indicate increased perceived work-related stress by employees, which in turn has negative effects on mental and general health. According to our results, a major cause can be found in a higher degree of interference between work and private life. The heterogeneity of home-based work settings and the limitations of an employer’s area of responsibility support personal resources such as self-efficacy as a key concept that mediates between work-related stress levels and health outcomes. At the same time, our results show that autonomy and remote work experience contribute to promoting self-efficacy. The present study contributes to a more holistic understanding of the benefits of employees’ self-efficacy as a critical personal resource that helps with lowering work-related stress and health risks.

## Figures and Tables

**Figure 1 ijerph-19-04955-f001:**
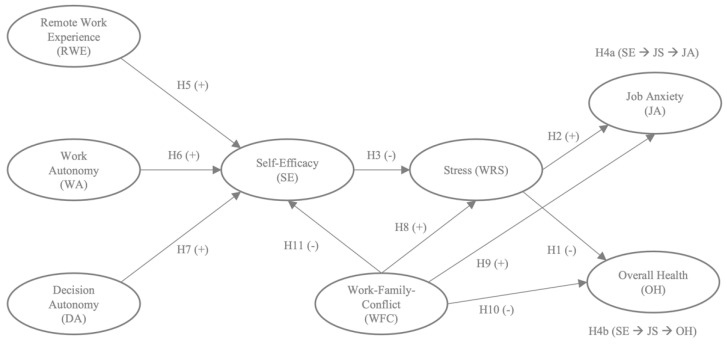
Conceptual Model.

**Table 1 ijerph-19-04955-t001:** Measurement model: reliability and validity.

	Mean	SD	CA	CR	R^2^	AVE	VIF
RWE	3.090	0.946	0.751	0.906	0.258	0.829	1.210
WA	3.985	0.849	0.818	0.893		0.736	1.804
DA	3.925	0.832	0.855	0.912		0.776	1.868
WFC	2.477	1.116	0.941	0.955		0.808	2.059
SE	4.078	0.671	0.872	0.902		0.535	1.580
WRS	2.563	0.999	0.870	0.905	0.156	0.657	2.244
JA	2.060	1.103	0.857	0.914	0.331	0.779	1.668

RWE—remote work experience; WA—work autonomy; DA—decision autonomy; WFC—work-family conflict; SE—self-efficacy; WRS—work-related stress; JA—job anxiety; SD—standard deviation; CR—composite reliability; CA—Cronbach’s alpha; R^2^—explained variance; AVE—average variance extracted; VIF—variance inflation factor.

**Table 2 ijerph-19-04955-t002:** Discriminant validity evaluation of the measurement model.

	RWE	WA	DA	WFC	SE	WRS	JA
RWE	0.910						
WA	0.088 **	0.858					
DA	0.172 **	0.649 **	0.881				
WFC	−0.039 **	−0.216 **	−0.153 **	0.899			
SE	0.330 **	0.377 **	0.419 **	−0.355 **	0.731		
WRS	−0.091 **	−0.237 **	−0.195 **	0.693 **	−0.386 **	0.811	
JA	0.109 **	−0.200 **	−0.193 **	0.525 **	−0.311 **	0.566 **	0.883

RWE—remote work experience; WA—work autonomy; DA—decision autonomy; WFC—work-family-conflict; SE—self-efficacy; WRS—work-related stress; JA—job anxiety; ** correlation sig. at 0.001; bold diagonal values—square root of the AVE values.

**Table 3 ijerph-19-04955-t003:** Structural model assessment.

Hypothesis	Coef Path	t-Value	*p*-Value	95% CI	f^2^
H1 (WRS→OH)	0.319	16.282	0.000	(0.280; 0.358)	0.048
H2 (WRS→JA)	0.396	24.278	0.000	(0.365; 0.427)	0.127
H3 (SE→WRS)	−0.164	13.989	0.000	(−0.187; −0.141)	0.047
H4a (SE→WRS→JA) *	−0.065 **	11.743	0.000	(−0.076; −0.054)	
H4b (SE→WRS→OH) *	−0.048 **	9.954	0.000	(−0.057; −0.039)	
H5 (RWE→SE)	0.215	18.784	0.000	(0.192; 0.237)	0.068
H6 (WA→SE)	0.153	8.672	0.000	(0.119; 0.187)	0.020
H7 (DA→SE)	0.260	15.049	0.000	(0.227; 0.294)	0.057
H8 (WFC→WRS)	0.634	62.723	0.000	(0.613; 0.653)	0.703
H9 (WFC→JA)	0.254	14.912	0.000	(0.220; 0.286)	0.052
H10 (WFC→OH)	−0.017	0.889	0.374	(−0.056; 0.020)	0.001
H11 (WFC→SE)	−0.281	22.964	0.000	(−0.304; −0.256)	0.112

* mediated effect; ** indirect effect; RWE—remote work experience; WA—work autonomy; DA—decision autonomy; WFC—work-family conflict; SE—self-efficacy; WRS—work-related stress; JA—job anxiety; Coef Path—Coefficient path; *p*-value—test of significance; f^2^—effect size; CI—confidence interval at 95%.

**Table 4 ijerph-19-04955-t004:** Explained variance of dependent constructs (R^2^) and predictive relevance (Q^2^).

Variable	R^2^	Q^2^
OH	0.072	0.071
JA	0.360	0.278
WRS	0.503	0.328
SE	0.332	0.175

Q^2^—predictive effect; R^2^—explained variance; OH—overall health; SE—self-efficacy; WRS—work-related stress; JA—job anxiety.

## Data Availability

Data will be available upon request to the corresponding author.

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
