# Peer review of "The Role of Self-Efficacy, Work-Related Autonomy and Work-Family Conflict on Employee’s Stress Level during Home-Based Remote Work in Germany"

_ijerph, 2022, doi:10.3390/ijerph19094955_

Round 1

Reviewer 1 Report

Dear Authors,

Thank you for the opportunity to read your work.

The theme chosen by the authors is not only relevant but necessary.

I would like to suggest some food for thought to support your work.

I completely agree that working from home was a panacea that was clearly uncovered by the pandemic.

Although I´m not able to read German, I endorse the inclusion of national references in the article, mainly to give context and show cultural differences.

Regarding Job Demand-Resource Model. The authors suggest that they are using this model as a framework. However, Bakker and Demerouti published a review about their work that I recommend the authors include (above the reference), and I would like to explain why.

From my point of view, the authors align with Bakker and Demerouti’s broad proposition but organise slightly different. Bakker and Demerouti suggest that job demands (in your case, work stress) and resources (autonomy and experience) would be moderated by personal demands (work-family conflict) and resources (self-efficacy) to explain the outcomes (health and anxiety).

In this sense, H3 is independent because work stress is not dependent on the person SE. From my point of view, it has important implications as, in some way, it is simple for the organisations to blame the person for the stress they feel because they present a low level of SE. Thus, the organisation removes the blame for the workers' stress levels.

There are situations and homes, especially when people must take care of other people (small children, dependent parents, disabled siblings, limited grandparents, sick uncles, etc.) in which there is no self-efficacy to cope, the person succumbs. In other words, it is a U-shaped relationship because if the person interprets that it is his or her fault that things work (or don't work) up to a certain point, it can be salutary. However, in a very fragile, stressful situation, a very high self-efficacy can work with a rebound effect, that is, get sicker.

There are cultural issues involved in caring for others (e.g. Masculinity-femininity cultural values), as well as in interpreting autonomy as a motivating variable (In the German Work Design scale validation, the “Entscheidungen” autonomy was one of the variables that differed the most between employee groups, showing that it is a relevant variable – although with a small sample), as well as a variable promoting well-being. Different results have been found about autonomy as a promoter of well-being in the pandemic context, most likely due to cultural issues. In other words, I believe that authors should clarify this cultural issue when advocating specific uses of variables and test the gender difference in the model.

Regarding the test and results:

The authors present a designed model but have not tested it in its entirety. Why didn't the authors test the whole model on a Path SEM model and present the model's modification index? In the absence of this entire test, the design of the model and its interpretation is limited.

Another possible interpretation of the absence of health outcomes (OH) could be the time that has elapsed. One year of pandemic did not allow time to impact people's physical health severely.

Overall, the work is robust, presents remarkable results, is very well written, is methodologically careful, and is well structured. It is a work with clear contributions to the literature. However, the strong influence of gender (pointed out by the authors themselves and feasible to be tested, given the robust sample - even if it is to say that there is no difference), the absence of testing of the entire model, as well as the possible interpretation of self-efficacy as worker "guilt" suggest caution in approving the paper as it stands and demands, in my opinion, a review of these points by the authors. 

I hope that these comments have been helpful.

Bakker, A. B., & Demerouti, E. (2017). Job demands-resources theory: Taking stock and looking forward. Journal of Occupational Health Psychology, 22(3), 273–285. https://doi.org/10.1037/ocp0000056

Author Response

Dear Reviewer 1,

we want to thank you for your appreciative hints and comments. Your review showed us important aspects and opened different perspectives on main issues. We adapted our manuscript accordingly. We apologize, if we did not go into the cultural aspect too much, as we did not see this within thiscope of our manuscript (mainly with regard to our data). We added a goodness of fit index (SRMR) to the results section and revised the discussion to minimize the “guilt”-aspect, which we did not intend to point out at all. Rather, we wanted to show readers what kind of resources employers should look for to support employees. We hope, that our answers will find your approval.

Note: Depending on the word proof setting the lines may be shifted. We referred to lines with markups turned on as the editors requested.

Author Response

Dear Reviewer 2,

we want to thank you for your appreciative hints and comments. Your review showed us important aspects and opened different perspectives on main issues. We adapted our manuscript accordingly. We hope that our answers will find your approval.

Note: Depending on the word proof setting the lines may be shifted. We referred to lines with markups turned on as the editors requested.

Reviewer 3 Report

It’s a very interesting, actual, and complex research. I have few remarks about this paper that I present below.

In the beginning of page 7, about “Measures” the authors didn´t identify the instruments they used, as they do for Minimum European Health Module I (MEHM1).  I remark that the names of those scales are in the bibliography. In this chapter they should include a brief description of each scale, and in the case of MEHM1 to show us the item used.

There are a lot of information from the Results and Hypothesis testing and from Discussion. By the opposite there are very generic and few “Practical Implications”. Could the authors present us some brief but specifics implications of these results?

In References, page 14, references 68 and 69 need the authors’ attention, it seems to be only one paper (also in page 7, about the MEHM1).

Author Response

Dear Reviewer 3,

we want to thank you for your appreciative hints and comments. Your review showed us important aspects for the method and practical implications parts. We adapted these parts accordingly. We hope, that our answers will find your approval.

Note: Depending on the word proof setting the lines may be shifted. We referred to lines with markups turned on as the editors requested.
